# Helicobacter Pylori: A Review of Current Treatment Options in Clinical Practice

**DOI:** 10.3390/life12122038

**Published:** 2022-12-06

**Authors:** Logan T. Roberts, Peter P. Issa, Evan S. Sinnathamby, Mallory Granier, Holly Mayeux, Treniece N. Eubanks, Kevin Malone, Shahab Ahmadzadeh, Elyse M. Cornett, Sahar Shekoohi, Alan D. Kaye

**Affiliations:** 1LSUHSC-New Orleans School of Medicine, 1901 Perdido Street, New Orleans, LA 70112, USA; 2Department of Anesthesiology, LSU Health Shreveport, 1501 Kings Highway, Shreveport, LA 71103, USA; 3LSU Health Shreveport, 1501 Kings Highway, Shreveport, LA 71103, USA; 4Departments of Anesthesiology and Pharmacology, Toxicology, and Neurosciences, LSU Health Shreveport, 1501 Kings Highway, Shreveport, LA 71103, USA

**Keywords:** *Helicobacter pylori*, treatment options, clinical practice

## Abstract

Background: When prescribing antibiotics, infection eradication rates, local resistance rates, and cost should be among the most essential considerations. *Helicobacter pylori* is among the most common infections worldwide, and it can lead to burdensome sequela for the patient and the healthcare system, without appropriate treatment. Due to constantly fluctuating resistance rates, regimens must be constantly assessed to ensure effectiveness. Methods: This was a narrative review. The sources for this review are as follows: searching on PubMed, Google Scholar, Medline, and ScienceDirect; using keywords: *Helicobacter pylori*, Treatment Options, Clinical Practice. Results: Multiple antibiotics are prescribed as part of the regimen to thwart high resistance rates. This can lead to unwanted adverse reactions and adherence issues, due to the amount and timing of medication administration, which also may contribute to resistance. Single-capsule combination capsules have reached the market to ease this concern, but brand-only may be problematic for patient affordability. Due to the previously mentioned factors, effectiveness and affordability must be equally considered. Conclusions: This review will utilize guidelines to discuss current treatment options and give cost considerations to elicit the most effective regimen for the patient.

## 1. Introduction

The gram-negative, spiral-shaped bacterium *Helicobacter pylori* is a widespread and prevalent opportunistic pathogen most commonly associated with gastritis, peptic ulcers, and various other gastrointestinal ailments [1]. There is robust evidence that *H. pylori* is acquired during childhood in most people, and is significantly determined by geography and quality of life [2]. Since the primary means of *H. pylori* infection are by oral–oral and fecal–oral transmission, risk factors include contaminated food or water supplies, contact with domestic animals, smoking, alcohol consumption, ill contacts, closely packed living situations, poor sanitation, poor hygiene, and most importantly, low socioeconomic status [3]. Noting the wide variety of risk factors and ease of transmission, it is no wonder that the global prevalence of *H. pylori* in one 2015 systemic review is estimated to be nearly 50%, with extremely high regional and national variability ranging from 18.9% to 87.7% [2]. A later 2018 meta-analysis by Alessia Savoldi et al. identified 178 studies in 65 countries, primarily in World Health Organization regions, and found: clarithromycin resistance of ≥15% in 11 of 15 countries (highest in Israel 47%, then France 43%), metronidazole resistance of ≥15% in 12 of 15 countries (highest in Israel 57%), levofloxacin resistance ≥15% in 5 of 15 countries (highest in Turkey 30%, then Belgium 18%) and negligible resistance to amoxicillin or tetracycline of <5% in most of the countries [4]. With such high global prevalence as an opportunistic pathogen, indications for *H. pylori* testing and subsequent eradication must be distinct to avoid unnecessary costs and risks. The American College of Gastroenterology provides clear guidelines for testing. The most common indications include the following: active peptic ulcer disease (PUD), low-grade gastric mucosa-associated lymphoid tissue (MALT) lymphoma, history of endoscopic resection for early gastric cancer, patients initiating chronic treatment with non-steroidal anti-inflammatory drugs (NSAIDs), unexplained iron deficiency anemia despite evaluation, and idiopathic thrombocytopenic purpura [5]. Should a test be indicated, biopsy-based tests, urea breath tests, stool-sample polymerase chain reaction (PCR) tests, and blood antibody tests are all potential options. Each option has benefits and drawbacks based on the clinical scenario, and decisions about testing are often made at the physician’s discretion [6]. Should *H. pylori* infection go untreated, severe complications such as gastritis, PUD, MALT lymphoma, stomach or esophageal cancer, and potentially idiopathic thrombocytopenic purpura may result [5] (see Figure 1). The cost of treatment for a symptomatic individual with *H. pylori* is relatively low, and proactive testing and eradication may decrease potential expenses related to an untreated infection’s extensive complications. In addition to decreasing costs, patients eradicated of *H. pylori* are less likely to require additional resources to treat their underlying dyspeptic symptoms, thereby decreasing demand on provider networks [7]. A 2018 Cochrane review of non-invasive diagnostic tests for *H. pylori* found that urea breath tests had high diagnostic accuracy, while serology and stool antigen tests had lower accuracy in detecting *H. pylori* infection; while there is an agreement of acceptable diagnostic studies, the decision on how to treat is not as simple [8]. Due to dynamic resistance rates, current treatment regimens often include multiple antibiotics with complex dosing strategies that burden the patient. To minimize these concerns, local resistance rates must constantly be monitored to ensure that the most effective strategies are implemented. A local repertoire of known cases of antibiotic resistance in the refractory treatment of *H. pylori* in collaboration with regional hospitals should be in place. However, this is hardly the case in the United States. Currently, with *H. pylori* there is an *H. pylori* monitoring program in Europe consisting of several nations collaborating and sending samples to a central repository. Such a program existed in the United States in the 1990s, dubbed the Surveillance of *H. pylori* Antimicrobial Resistance Partnership (SHARP) program, that tracked antibiotic resistance; however, this program was short-lived [9]. This review will evaluate recommended treatment options, resistance rates, best practice clinical decision-making, and cost considerations.

## 2. Materials and Methods

This was a narrative review. The sources for this review are as follows: searching on PubMed, Google Scholar, Medline, and ScienceDirect, and using keywords: Helicobacter Pylori, Treatment Options, Clinical Practice.

## 3. *H. pylori* Overview

*H. pylori* were discovered as spiral bacteria in the stomach of dogs by Giulio Bizzozero in 1892. Since they are Campylobacter-like spiral bacteria, Barry Marshall and Robin Warren dubbed them Campylobacter pyloridis in 1983. Goodwin et al. designated them Helicobacter pylori in 1989 because of their helical form, and prevalence in the pyloric area of the stomach. *H. pylori* are a 0.5–1 m wide, 2–4 m long, S-shaped, short helical, Gram-negative bacteria that infect more than half of the world’s population [10].

*H. pylori* are a short helical, S-shaped, Gram-negative bacteria measuring 0.5–1 m in width and 2–4 m in length. They are particularly prevalent in the pyloric area of the stomach, where they cause persistent gastric infection. It is believed that more than half of the world’s population is infected with these bacteria. The specific modes of transmission and infection with *H. pylori* are yet unknown, but the feces-to-mouth and mouth-to-mouth pathways via water or food consumption are believed to be quite common [10] (Figure 2 and Figure 3).

The severity of *H. pylori*-caused gastric atrophy and gastric cancer has been the subject of growing study attention over the past three decades, particularly in terms of pathogenicity, microbial activity, genetic predisposition, and clinical therapies. Studies have revealed a connection between *H. pylori* infection and malabsorption of important micronutrients, and *H. pylori* infection may influence the prevalence of malnutrition in some high-risk populations. Dietary factors may play a significant role in *H. pylori* infection, and a balanced diet, particularly with a high consumption of fruits and vegetables and low consumption of processed salty foods, has a protective effect against the consequences of *H. pylori* infection [10]. Figure 4.

The combination of drug loading of nano materials and natural drug loading may have a better antibacterial effect in future clinical studies. Biotherapeutics derived from microorganisms are crucial for combating infections such as *H. pylori*, but these biotherapeutics must be alive at the time of administration to be effective. Numerous potentially therapeutic species are anaerobes and, as a result, their generation is almost impossible due to the low efficacy of present protective techniques.

Inspired by the features of cells in living animals, these new hybrids, comprised of living cells and abiotic materials with diverse forms and functions, can increase cell stability and allow for the introduction of novel activities into living cells. Numerous applications, such as bioelectronics, cell protection, cell treatment, and biocatalysis, have a significant deal of potential for single-cell nanoshells [11].

For example, Bacteroides thetaiotaomicron is a self-assembling cellular coating [12]. Even in the absence of conventional cryoprotectants, this coating exhibits resistance to harsh processing conditions and oxygen exposure. This innovation will expand the variety of microorganisms that can be produced in a stable manner, and promote the development of new strains of interest by ensuring their survival after manufacturing. Reversible cell encapsulation, made possible by the presence or absence of glucose, is another emerging development that significantly increases cell survival in a variety of hostile environments, without interfering with the cells’ natural growth [13].

## 4. Etiology, Epidemiology, Pathophysiology

*H. pylori* can be transmitted by the fecal–oral, gastric–oral, oral–oral, and sexual pathways. Lower socioeconomic status is a significant risk factor for a higher infection prevalence [14]. The prevalence of *H. pylori* varies across the globe, with a 5% prevalence in children younger than ten years old in the United States. It is more prevalent in the Hispanic and African American populations than the White population [14].

In *H. pylori* infection, four key factors contribute to the development of clinical illnesses such as gastritis and ulcer. First, the urease activity of *H. pylori* serves a crucial function in neutralizing the stomach’s acidic environment. Second, the *H. pylori* bacterium moves toward the host gastric epithelial cells via flagella-mediated motility. The subsequent interaction between bacterial adhesins and host cell receptors results in effective colonization and sustained infection. In addition, *H. pylori* produce numerous effector proteins/toxins, such as cytotoxin-associated gene A (Cag A) and vacuolating cytotoxin A (VacA), that cause host tissue damage. *H. pylori* gastritis is characterized by both acute and chronic inflammation due to the stimulation of eosinophils, neutrophils, mast cells, and dendritic cells. Additionally, the gastric epithelial layer secretes chemokines to trigger innate immunity and activates neutrophils, which further damage the host tissue, resulting in the establishment of gastritis and ulcer [14].

## 5. Antibiotics Overview

### 5.1. Clarithromycin

Clarithromycin is a broad-spectrum antimicrobial (macrolide class) known for covering atypical organisms. Clarithromycin is one of the five approved macrolides in the United States. It is well-known for its use against *H. pylori* as it remains a staple therapeutic in the triple therapy regimen. Similarly to others in the macrolide class, clarithromycin is a bacteriostatic antibiotic that reversibly binds the large 50S subunit of bacterial ribosomes at the 23S ribosomal RNA (rRNA) to inhibit protein synthesis [15,16]. 

Though widely considered the most effective antimicrobial agent for treating and eradicating *H. pylori*, the use of clarithromycin is not without significant side effects [17,18]. Adverse events associated with using clarithromycin are cited as high as 86%, most of which are gastrointestinal or taste-related [19]. The most common adverse event is taste perversion, reported with a frequency of 58% in a randomized, double-blind trial. Gastrointestinal side effects, including nausea, vomiting, diarrhea, and abdominal discomfort, are also especially common [20]. Such adverse events tend to present more commonly in children [21]. As a known cytochrome P450 (CYP) 3A4 enzyme inhibitor, clarithromycin can influence the pharmacokinetics of other drugs, and consequently induce hepatotoxicity. In addition, clinicians are recommended to pay close attention to a patient’s QT interval and corresponding electrolytes, specifically Potassium and Magnesium, as clarithromycin can potentially prolong the QT interval. Clarithromycin resistance, once thought to be at or less than 5% in most countries, is becoming increasingly more prevalent worldwide. This increasing resistance may be due to the usage of clarithromycin in otorhinolaryngology, respiratory, in pediatrics, and due to its increased prescribing for the treatment of *H. pylori* [22,23,24]. Biochemically, resistance has been seen with the inhibition of binding between clarithromycin and the ribosomal subunit dedicated to specific antibiotic-related protein synthesis. Resistance has been attributed to modifications to the efflux pump system [25]. Accordingly, clarithromycin resistance rates have been reported to be as high as 27% [26].

### 5.2. Amoxicillin

Amoxicillin is an oral aminopenicillin within the beta-lactam class of antibiotics. Amoxicillin has garnered favorability because, in addition to gram-positive coverage typical of natural penicillins, it also has utility against gram-negative pathogens, including Haemophilus, Neisseria, Proteus, and *E. coli* [27]. Furthermore, the use of beta-lactamase inhibitors (e.g., clavulanate) can increase amoxicillin’s effectiveness against resistant gram-negative and methicillin-susceptible Staphylococcus aureus (MSSA) [28,29]. Commonly used to treat infections of the respiratory tract, urinary system, and ear, amoxicillin is also frequently used to treat *H. pylori* as it is one of the medications in the triple therapy regimen [30,31]. Beta-lactams bind penicillin-binding proteins, inhibiting transpeptidation, and consequently inducing autolytic destruction of the bacterial cell wall [32]. Minimum inhibitory concentrations of amoxicillin against *H. pylori* have been reported in one study at 0.0156–256 mg/L (MIC50 0.125 mg/L, MIC90 4 mg/L) in patients who had not previously received *H. pylori* treatment [33]. Treatment regimens vary based on several factors. The usual dose for patients without risk factors for macrolide resistance is 1 g twice daily in combination with clarithromycin 500 mg twice daily [5]. Common side effects of amoxicillin use include gastrointestinal symptoms (e.g., diarrhea, nausea, vomiting, abdominal discomfort), nephrotoxicity (e.g., crystalluria, nephritis), hepatotoxicity, and hypersensitivity reactions type I-IV. Serious complications may manifest in rare instances, such as Steven Johnson Syndrome or seizures [30,34]. Accordingly, patient history of a previous hypersensitivity reaction to penicillins is a contraindication for amoxicillin use [35]. 

### 5.3. Bismuth Subsalicylate

Bismuth subsalicylate is an antacid, antidiarrheal, anti-inflammatory, and bactericidal agent which is widely known as the active ingredient in Pepto-Bismol [36,37]. Bismuth subsalicylate is fragmented in the stomach into salicylate and bismuth, which is minimally absorbed to act as a bactericidal agent. In the alimentary canal, bismuth reduces inflammation, minimizes fluid excretion, and prevents bacterial adhesion [37,38,39]. Bismuth subsalicylate is generally well tolerated, although common side effects include diarrhea, nausea, bitter taste, and dark stools [40]. Rarely, patients may present with a blackened tongue, mood changes, or neurotoxicity [41,42]. Contraindications for bismuth subsalicylate use include patients with gastrointestinal ulcers, bleeding problems, black stools before administration, or already consuming medications high in salicylate (such as anticoagulants, methotrexate, etc.). In addition, adolescents with flu-like symptoms, children aged 12 years or younger, or patients with a history of an allergic reaction to subsalicylates are not recommended to consume bismuth subsalicylate [43,44]. A maximum intake of 4200 mg within 24 h is recommended.

### 5.4. Metronidazole

Metronidazole is a narrow-spectrum synthetic antimicrobial of the nitroimidazole class. The reduction of metronidazole, following its entry into the bacterium, is thought to produce an intermediate responsible for cytotoxic and antimicrobial effects [45,46]. Though side effects are uncommon, metronidazole may cause vomiting, nausea, diarrhea, and abdominal pain. Patients taking metronidazole in oral form also complain of a metallic taste. More serious and rarer adverse events include numbness, peripheral neuropathy, and seizures [47]. Significantly, alcohol can interact with metronidazole in a disulfiram-like reaction that typically presents with flushing, cramping, vomiting, tachycardia, and palpitations [48,49]. To avoid this, patients should refrain from alcohol use while taking metronidazole, and 72 h following the last dose of metronidazole. Notably, prescribing should be cautioned in patients with central nervous system abnormalities as it may exacerbate symptoms [50]. Resistance varies greatly depending on regional areas [25]. Resistance is primarily due to mutations involving the gene rdxA; complex genetic events (deletions of transposons, missense mutations, frameshift mutations, and insertions) can be simultaneously present with mutation of this gene, of which these genetic events are caused by metronidazole’s production of DNA-damaging compounds [25].

### 5.5. Tetracycline/Doxycycline

Since the 1950s, tetracycline is a broad-spectrum bacteriostatic agent that inhibits protein synthesis by binding to the 30S subunit, and effectively limiting hydrogen bond formation between amino acids [51,52]. In addition to its function as an antibacterial agent, tetracycline has antiparasitic activity, inhibiting the growth of Plasmodium falciparum, Giardia lamblia, and Trichomonas vaginalis [53]. Tetracycline has long been reported to interfere with bone mineralization and calcification; consequently, its use is contraindicated in patients under the age of 8 years [53]. Importantly, the risk of maternal hepatotoxicity renders tetracycline a potential teratogen, and its use should be avoided in pregnant and breastfeeding women [54]. Unfortunately, tetracycline-resistant bacteria were discovered shortly after its introduction. While currently tetracycline resistance is uncommon, and resistance to levofloxacin, clarithromycin, and metronidazole are predominant, resistance to tetracycline remains a considerable concern in the future as more bacteria become resistant to antibiotics [55,55,56]. Tetracycline exerts its effect on the 30S subunit of the ribosome of *H. pylori*, and blocks the binding RNA; the resistance of *H. pylori* to tetracycline is conferred by mutations in the 16S rRNA, allowing RNA to bind [25].

### 5.6. Levofloxacin

Levofloxacin is a broad-spectrum antibiotic of the fluoroquinolone class. Known for its concentration-dependent bactericidal activity, high-dose, short-course levofloxacin has become an attractive therapy that maximizes the chance of regimen course completion, while minimizing the risk for resistance development [57]. Levofloxacin acts on gram-positive and gram-negative bacteria by inhibiting DNA replication, specifically DNA gyrase and topoisomerase, and is accordingly considered a bactericidal [58]. From a safety perspective, it is important to note that levofloxacin and the fluoroquinolone class possess a black box warning label for their association with tendinopathy and tendon rupture, central nervous system effects, and peripheral neuropathy [59]. Mild and common side effects of levofloxacin use include gastrointestinal symptoms, diarrhea, abdominal pain, and central nervous system-related disturbances such as dizziness, headache, and insomnia [60,61]. Other potential complications include psychiatric disturbances, such as agitation, disorientation, suicide ideation, QT prolongation, drug absorption interactions, hyper- and hypoglycemia, and photosensitivity. Importantly, levofloxacin is not recommended for use in Myasthenia gravis patients, considering both musculoskeletal and central nervous system-related disturbances [62]. In addition, levofloxacin use is not recommended for patients aged 18 years or younger, pregnant, or nursing women [61]. Recently, the resistance rate of levofloxacin has been increasing, with recent articles suggesting using newer fluoroquinolones such as sitafloxacin [63].

## 6. Acid Suppressants Overview

### 6.1. Proton Pump Inhibitors (PPIs)

Proton pump inhibitors (PPIs) are a class of medications that effectively reduce gastric acid production by irreversibly inhibiting the luminal H+/K+ ATPase of parietal cells within the stomach [64]. Accordingly, PPIs commonly treat gastroesophageal reflux disease, and peptic ulcer disease. They are also vital in *H. pylori* eradication as they are commonly included in triple therapy regimens [65,66]. Considering *H. pylori’s* preference for an acidic environment, raising gastric mucosal pH with PPIs can hinder bacterial growth, and allow gastric ulcer recovery. PPIs are also thought to allow antibiotics to concentrate in the stomach, further strengthening the efficacy of the triple therapy regime [67]. Though typically well-tolerated, PPIs can cause abdominal discomfort, dizziness, and nausea [68,69]. In addition, PPI use is commonly associated with an increased risk of enteric infections [69]. Considering drug metabolism, lower doses are generally recommended for patients with hepatic disease [70].

### 6.2. Vonoprazan

Vonoprazan belongs to a new class of acid-suppressant medications known as K-competitive acid blockers. K-competitive acid blockers, also called acid pump antagonists, exert their effect by minimizing potassium availability in the lumen, reducing K+/H+ ATPase cotransport, and consequently decreasing gastric acid production [71,72]. Unlike PPIs, vonoprazan binds reversibly, and has a longer half-life [73,74]. Importantly, vonoprazan was FDA-approved in 2022 to be used in triple therapy regimens to treat *H. pylori* [75,76]. Common side effects of vonoprazan use include abnormal hepatic function, rash, and drug eruption. Though functioning similarly to PPIs, vonoprazan demonstrates a slightly different safety profile with an increased risk for enterocolitis hemorrhage, and drug eruption [77].

## 7. First-Line Therapies

### 7.1. Bismuth Quadruple Therapy

The bismuth quadruple therapy (BQT) is the recommended first-line initial treatment option when areas are exhibiting high levels (>15%) of clarithromycin resistance, and low-level dual clarithromycin and metronidazole resistance (<15%) [78]. It is also the recommended first-line therapy in patients with recent macrolide exposure, or who are allergic to penicillin [5]. The BQT includes bismuth subsalicylate, metronidazole, tetracycline, and a PPI [5]. It consists of 300 to 524 mg of bismuth subsalicylate four times daily, 500 mg of metronidazole 3 to 4 times daily or 250 mg 4 times daily, and 500 mg of tetracycline hydrochloride four times daily with a standard-dose PPI [79].

Depending on insurance coverage and affordability, the BQT regimen can be given as an FDA-approved three-in-one capsule, plus a PPI. These combination capsules have been shown to improve compliance and tolerability. In contrast, bismuth, PPIs, tetracycline, and metronidazole prescribed as separate drugs are not FDA-approved. Pylera, a fixed-dose capsule containing bismuth subcitrate, tetracycline, and metronidazole, combined with a PPI for ten days, and Helidac, a co-packaged product containing bismuth subsalicylate, tetracycline, and metronidazole, combined with a PPI for 14 days, are FDA-approved treatment regimens. The duration of BQT should be between 10 and 14 days [5]. ITT analysis shows an eradication rate of 88.5% when BQT is taken as recommended [80]. This regimen’s failure rate has been around 10% in North America [81,82].

### 7.2. Clarithromycin Triple Therapy

Clarithromycin triple therapy consists of a standard dose of PPI, clarithromycin 500 mg, and amoxicillin 1 g, all taken twice a day, or metronidazole 500 mg three times daily [5]. Metronidazole 500 mg three times daily is used as a substitute when the patient is allergic to penicillin, and it is recommended to be given over 14 days [5]. Recent studies have shown that the clarithromycin triple therapy eradicates about 77% of the population in the United States. Still, its success depends highly on the local level of clarithromycin resistance. Although the use of clarithromycin remains exceedingly successful in susceptible strains, it should be avoided when resistance rates exceed 15%. Data suggested that in 2017 when the ACG Guidance was released, North American clarithromycin resistance rates were between 15 and 20% [83]. 

### 7.3. Concomitant Therapy

The so-called “Concomitant Therapy” consists of a standard dose of PPI, clarithromycin 500 mg, and amoxicillin 1 g with the addition of metronidazole 500 mg or tinidazole 500 mg, all taken twice a day [5]. It should be considered in patients intolerant of bismuth. Concomitant therapy over 14 days yields the highest cure rates [84]. In a recent quasi-experimental comparative study, concomitant therapy was shown to have an eradication rate of 84%, compared to a 77% eradication rate for triple therapy [85]. Because of its efficacy, it can be recommended as first-line therapy [5]. Clarithromycin resistance may reduce the effect of concomitant therapy, but this is to a lesser degree than with the clarithromycin triple therapy [86]. If used, therapy should last for a duration of 10 to 14 days.

### 7.4. Sequential Therapy

Clarithromycin-based sequential therapy consists of a standard dose of PPI plus amoxicillin 1 g twice daily for five days [5]. This is followed by a PPI, clarithromycin 500 mg, and either metronidazole or tinidazole at a dose of 500 mg twice daily for an additional five days. While it is comparable to clarithromycin-based triple therapy, its complexity detracts from its viability as a first-line agent. Sequential therapy has been shown to have an eradication rate of 84.3% [5]. It has not been shown to have superior outcomes to either a 14-day clarithromycin-based triple therapy or a 10–14-day bismuth quadruple therapy. Tolerability and compliance are similar to the clarithromycin-based triple therapy [80]. Due to the complexity of sequential therapy, and the lack of evidence of superiority compared to 14-day clarithromycin triple therapy, clarithromycin-containing sequential therapy has not been endorsed by guidelines in North America as a first-line treatment [87].

### 7.5. Hybrid Therapy

Hybrid therapy is a combined sequential and concomitant therapy. It consists of a standard dose of PPI plus Amoxicillin 1 g taken twice a day for seven days, followed by PPI, Amoxicillin, Clarithromycin 500 mg, and either metronidazole or tinidazole 500 mg taken twice a day for an additional seven days [5]. While there are a lack of data supporting hybrid therapy’s effectiveness in North America, it has been shown to have high cure rates in international studies. Several international studies support using hybrid therapy as an alternative to clarithromycin triple therapy. A study by Wang et al. showed that hybrid therapy’s eradication rate was 88.6% [88]. The tolerability of hybrid therapy is similar to that of clarithromycin-based triple therapy [80]. There is also no significant difference in compliance, efficacy, or tolerability between the hybrid and concomitant therapies [88]. 

## 8. Common Substitutions and Cautions 

### 8.1. Penicillin Allergy

Amoxicillin is used in many first-line therapies to treat an *H. pylori* infection [4]. Several therapies do not contain amoxicillin, such as the bismuth quadruple therapy. The clarithromycin-based triple therapy can also be used if metronidazole is substituted for penicillin.

### 8.2. Alternatives to Clarithromycin

Clarithromycin-based triple therapy may be a first-line therapy to treat *H. pylori* with low regional levels of resistance. This therapy is not recommended when clarithromycin resistance rates are >15–20% [89]. Other therapies such as the bismuth quadruple or sequential therapies are suggested [9]. Levofloxacin-based therapies are also used as an alternative to clarithromycin. Levofloxacin has been shown to potentially have an eradication rate of >90%, especially where there is low resistance to levofloxacin [90]. The resistance to quinolones is rising; however, levofloxacin is discouraged as a first-line treatment [86]. 

### 8.3. Metronidazole and Alcohol Use

Metronidazole is contraindicated in patients who recently consumed alcohol or products that contain propylene glycol. Patients should avoid consuming alcohol until three days after metronidazole-containing therapy. There have been reports of a disulfiram reaction occurring in patients consuming alcohol while being administered metronidazole. Typically, disulfiram reactions present with flushing, nausea, vomiting, tachycardia, and palpitations [91,92].

### 8.4. Tetracyclines and Pregnancy

Tetracycline is contraindicated in pregnant women due to a risk of hepatotoxicity in the mother, and permanent discoloration of the teeth in the fetus [93]. There is also a risk of impaired fetal bone growth development [93]. For this reason, bismuth quadruple therapy is not advised in pregnant patients. Other first-line therapies, such as standard triple therapy, are recommended in their place. 

### 8.5. Follow Up Eradication Confirmation

It is recommended that patients receive post-treatment eradication testing at least four weeks after the completion of antibiotic therapy [94]. PPIs should be withheld for at least one to two weeks before testing to avoid false negatives [95]. Common eradication tests include a urea breath test, fecal antigen testing, or histology if upper endoscopy is performed. Serology testing is not usually recommended for eradication confirmation. One caveat is that it has been proposed to increase the interval of stool antigen testing from 4 to 6–8 weeks to decrease the chance of false positives [96]. If eradication testing is negative, additional testing is not typically needed. If eradication testing is positive, additional eradication therapy is warranted.

## 9. Second-Line Agents for Treatment Failure 

### 9.1. Suggested Approach

With increasing *H. pylori* strain diversity, many first-line treatments can be ineffective at eradication. Patients require a second-line salvage therapy regimen to be administered when a refractory infection occurs or when there is a persistent positive non-serological test occurring up to four weeks after the first-line treatments are administered. When choosing a second-line therapy, patient allergies, first-line therapies previously used, local geographic resistance rates, and sensitivities should be considered. Due to low global resistance rates to amoxicillin, this antibiotic may be reused in second-line therapy regimens even if it was initially used in first-line treatment. Other previously used regimens should be avoided. Suppose there is a failure of first-line therapy in a penicillin-allergic patient. In that case, it is advised that the patient receives allergy testing to identify if they have a true penicillin allergy [5]. Numerous studies show that about 5–10% of Americans state that they are allergic to penicillin, though over 90% of these patients have a negative skin test and can tolerate penicillin well. Sensitivity testing is a major source of overcoming treatment failure due to antimicrobial resistance, with various strains of *H. pylori* displaying some forms of resistance to current therapies throughout the world; determining the sensitivities of strains before therapy administration is a crucial way to combat the increase in resistance [97]. Worldwide antibiotic resistance has been increasing at an alarming rate. 

### 9.2. Bismuth Quadruple Therapy 

Bismuth quadruple therapy (BQT), discussed in “initial antibiotic selection,” may be used for 14 days as a second-line regimen in treatment if previously unused as the first-line option [5]. Many alternative forms of bismuth-containing therapies are quadruple therapy variations of levofloxacin- and rifabutin-based triple therapy regimens, and are discussed later in this section.

### 9.3. Levofloxacin-Based Therapy

Levofloxacin-based triple therapy, consisting of levofloxacin, amoxicillin, and a proton pump inhibitor, is typically administered over 10 to 14 days as a second-line regimen. The dosages of each component are administered as follows: 500 mg of levofloxacin daily, 750 mg of amoxicillin three times daily, and a standard dose of a proton pump inhibitor twice daily. Levofloxacin-based therapy regimens are not indicated as second-line treatments in regions of the world with resistance rates greater than 15%, unless the strain has known sensitivity to the drug. This is due to levofloxacin experiencing increasing rates of primary and secondary resistance. Primary resistance rates ranging from 11 to 30% have been documented from data from over 50,000 patients across 45 countries, and those rates rise to 19–30% when considering secondary resistance after unsuccessful *H. pylori* treatment [97].

Levofloxacin-based quadruple therapies exist in addition to their triple therapy counterparts. Four popular regimens exist that all center around levofloxacin, with three of them containing bismuth as a core component.

LOAD therapy, containing levofloxacin, omeprazole, nitazoxanide, and doxycycline, is a novel treatment regimen for *H. pylori* eradication. Therapy lasts 7–10 days compared to the traditional 14 days of the standard triple therapy, with dosages consisting of 250 mg of levofloxacin once daily, 40 mg of omeprazole twice daily, 500 mg of nitazoxanide twice daily, and 100 mg of doxycycline once daily. Compared to the standard triple therapy for *H. pylori* eradication, eradication rates were significantly greater using a LOAD regimen (82.75% vs. 60.26%) [98]. Even with higher eradication rates, treatment is still considered suboptimal with LOAD regimens, likely due to rising quinolone resistance and shorter treatment duration. 

The three levofloxacin-based quadruple therapies, including bismuth as the main component, are the PBLA, PBLT, and PBLM regimens. If a patient has no known allergy to penicillin, PBLA therapy is the recommended regimen; however, if the patient is allergic to penicillin, then PBLT or PBLM regimens are recommended. 

PBLA therapy, consisting of a proton pump inhibitor, bismuth, levofloxacin, and amoxicillin, is an alternative form of levofloxacin quadruple therapy containing bismuth. With therapy lasting 7–10 days, similar to the LOAD regimen, concerns of suboptimal efficacy still arise when this treatment is used. Dosages consist of an individual’s standard dose of a proton pump inhibitor twice daily, 120 mg of bismuth subcitrate four times daily, 500 mg of levofloxacin once daily, and 1 g of amoxicillin twice daily. PBLT therapy, consisting of the same components as PBLA therapy, except for substituting tetracycline for amoxicillin, and PBLM therapy, substituting amoxicillin with metronidazole, display similar therapy durations to the previously mentioned LOAD and PBLA regimens. Resistance rates among the three varying antibiotics used in these regimens vary greatly, with metronidazole experiencing resistance in 30–65% of secondary treatment cases, and amoxicillin and tetracycline occurring in less than 5% of strains undergoing secondary treatment [97].

Levofloxacin sequential therapy is another rescue therapy used commonly in treating *H. pylori* infections. It consists of administration of 1 g amoxicillin and a standard dose of a proton pump inhibitor twice daily over 5–7 days, immediately followed by administration of a proton pump inhibitor, 500 mg of levofloxacin four times daily, and 500 mg of metronidazole three times daily for another 5–7 days. This treatment regimen has been shown previously to eradicate the infection in 90% of patients, and is considered a significantly better regimen than the standard triple therapy for *H. pylori* eradication [99].

### 9.4. High-Dose Dual Therapy

High-dose dual therapy, consisting of a proton pump inhibitor and 750 mg of amoxicillin given four times a day or 1 g three times daily over 14 days, is a superior regimen to standard rescue therapies in *H. pylori* infections. In one study, infections were eradicated in 95.3% of patients given the high-dose dual therapy regimen [100]. This can be particularly useful in patients with dual clarithromycin and levofloxacin-resistant strains.

### 9.5. Rifabutin Triple Therapy

Rifabutin triple therapy, a regimen consisting of 750 mg of amoxicillin three times daily, a proton pump inhibitor twice daily, and 300 mg of rifabutin given once daily over 14 days, has been shown to exhibit eradication rates of 83.8% versus 57.7% found in amoxicillin and proton pump inhibitor administered alone [101]. The significant difference in efficacy is most likely attributable to the increased prevalence of amoxicillin resistance in *H. pylori* strains, similar to the differences in the efficacy of other previously mentioned regimens.

### 9.6. Clarithromycin-Based therapy

In clarithromycin-based therapy containing a proton pump inhibitor, bismuth, and tetracycline, eradication rates in non-resistant strains of *H. pylori* have been shown to be greater than 95% successful [102]. Clarithromycin should be considered when macrolide resistance is not a concern, and the local resistance rates are less than 15%. While clarithromycin is still considered for use in rescue therapy following unsuccessful *H. pylori* eradication, the high prevalence of resistance in *H. pylori* strains to the antimicrobial often indicates the use of other drug classes in treatment [97].

### 9.7. Factors Associated with Treatment Failure

While many factors are associated with the failure of *H. pylori* eradication, the main contributors are patient noncompliance and increased antimicrobial resistance, especially to quinolones and macrolides. This resistance has also been shown to be both primary and secondary in various *H. pylori* strains throughout different regions of the world [97]. When considering that many antimicrobials function at a higher pH than stomach acid allows, inadequate acid suppression during treatment is also a major source of failure to eradicate infections. This possible point of failure focuses on the cytochrome P450 gene CYP2C19. This gene is responsible for the majority of the metabolism of early-generation proton pump inhibitors. It has been shown to have many polymorphisms affecting an individual’s drug clearance rate. Those with polymorphisms that increase metabolism have been associated with higher rates of eradication failure, despite exposure to susceptible antibiotics [97]. Although genetic testing is not recommended, it can provide insight into potential antibiotic failures in treatment. For those with inadequate acid suppression, alternative therapies include high-dose PPIs; potent PPIs that rely less on CYP2C19 metabolism, such as esomeprazole; or substituting for vonoprazan with either amoxicillin alone, or the combination of clarithromycin and amoxicillin, may offer improved acid suppression. 

Many problem-solving avenues may be considered to reduce the burden of *H. pylori* treatment failure. Patient compliance with therapy regimens has decreased as the complexity of treatment, such as the high pill burden and side effects, increases [103]. The invention of novel antimicrobial agents specific to *H. pylori* and the possible repurposing of already available antimicrobials that may have secondary mechanisms affecting *H. pylori* strains are also being explored. 

### 9.8. Probiotics

Probiotics, live microorganisms that can be administered as therapy, may have a future role in eradicating *H. pylori* by alleviating antibiotics’ side effects. The most commonly used probiotic organisms include Lactobacillus and Bifidobacterium, two Gram (+) organisms. Posed benefits to the host include promoting gut maturation/integrity, pathogen antagonism, and immune system modulation. A primary characteristic of these bacteria is their ability to anaerobically digest saccharides to produce lactic acid, a cellular product that inhibits *H. pylori.* Their beneficial effects come from nonimmunological mechanisms such as strengthening the mucosal barrier of the GI tract, as well as secretion of antimicrobial substances such as short-chain fatty acids. They also exhibit beneficial effects through immunological mechanisms, such as releasing anti-inflammatory cytokine secretion that reduces inflammation, and gastric acid production [102,103,104]. Several studies show the beneficial effects of probiotics when combating *H. pylori*. The indications for use are a supplement to first-line therapies that may help alleviate some treatment-related side effects [90]. 

## 10. Cost Considerations

Patient affordability should always be considered when constructing an appropriate treatment regimen. Even the most appropriate medications are ineffective if they never reach the patient. Table 1 was created, using GoodRx, to calculate the cheapest cost to obtain 14 days of each regimen without insurance in the southeastern United States [105]. It should be noted that GoodRx prices are a cross-sectional regional estimation that varies depending on local pharmacies. Many generic options yield affordable copays for those with insurance, except for the newer brand combination capsules. If considering a combination capsule to ease the pill burden, copays can still be excessive despite insurance coverage. This scenario can be circumvented by utilizing manufacturer coupon cards to reduce copays. If a generic is to be allowed onto the market, these prices may become more affordable for the patient, and improve patient adherence. Overall, consideration of insurance formulas, deductibles, a patient’s willingness to pay, and possession of an insurance policy can improve patient outcomes by initiating cost-effective regimens sooner. To the author’s knowledge, there are no recent cost comparisons of *H. pylori* treatment in the United States; further investigations are warranted, although out of the scope of this paper (see Table 2 and Table 3).

## 11. Conclusions

It has been proven that receiving treatment to eradicate *H. pylori* infections is more beneficial than not, due to the long-term complications associated with non-treatment, such as gastritis, gastric ulcerations, and malignancies. However, choosing the appropriate treatment regimen is immensely dynamic, due to high resistant rates, prescription costs, side effects, and patient non-adherence. Due to high clarithromycin resistance rates, bismuth quadruple therapy can often be the most appropriate initial antibiotic selected for eradication. If resistance is not a local problem, clarithromycin triple therapy is still a viable first-line option due to its inexpensiveness, and well-established effectiveness. Combination capsules circumvent patient adherence concerns but should only be considered contingent upon patient affordability. If side effects create a barrier to adherence, probiotics should be considered as an addition to a regimen to ease the side effect burden. When deciding between other second-line regimens, it is important to make a patient-specific selection based on allergies, effectiveness, affordability, side effects, and ease of administration. As current regimens encounter fluctuations in resistance patterns, repurposed medications such as rifabutin may pose substantial benefits in eradicating infections in treatment-failure patients.

## Figures and Tables

**Figure 1 life-12-02038-f001:**
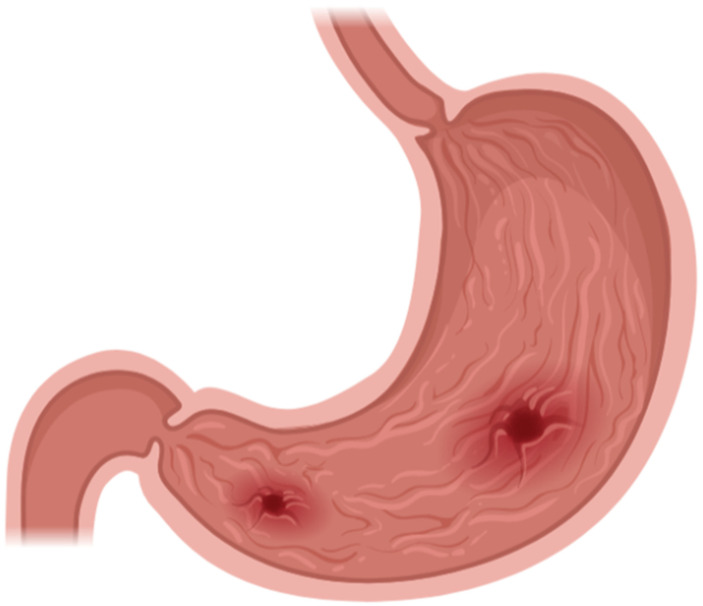
Drawing of *H. pylori*-induced gastritis. Should *H. pylori* infection go untreated, severe complications such as gastritis, PUD, MALT lymphoma, stomach or esophageal cancer, and potentially idiopathic thrombocytopenic purpura may result.

**Figure 2 life-12-02038-f002:**
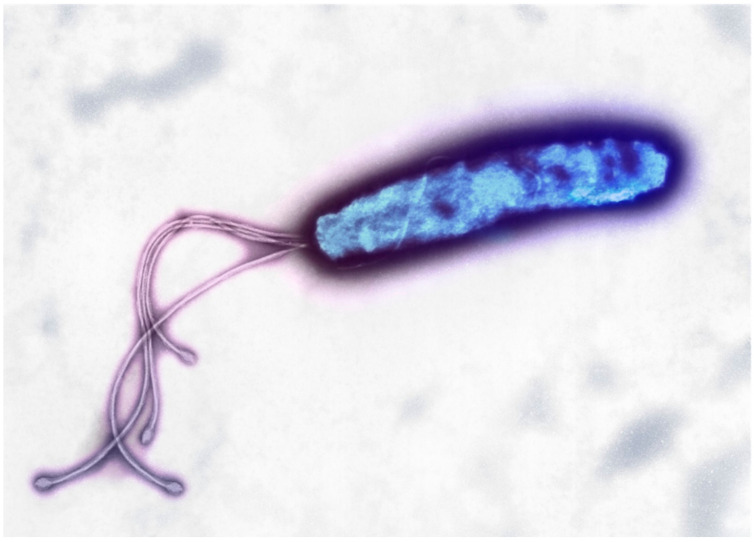
*H. pylori* bacterium, colored transmission electron micrograph (TEM). Image paid for by authors; stock photo from Canva.

**Figure 3 life-12-02038-f003:**
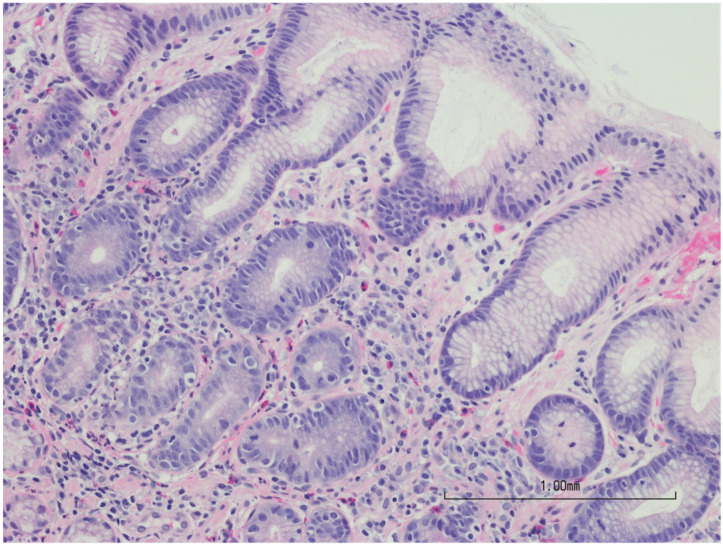
Stained microscopic image of stomach biopsy; reactive epithelium from *H. pylori*-induced gastritis. Image paid for by authors; stock photo from Canva.

**Figure 4 life-12-02038-f004:**
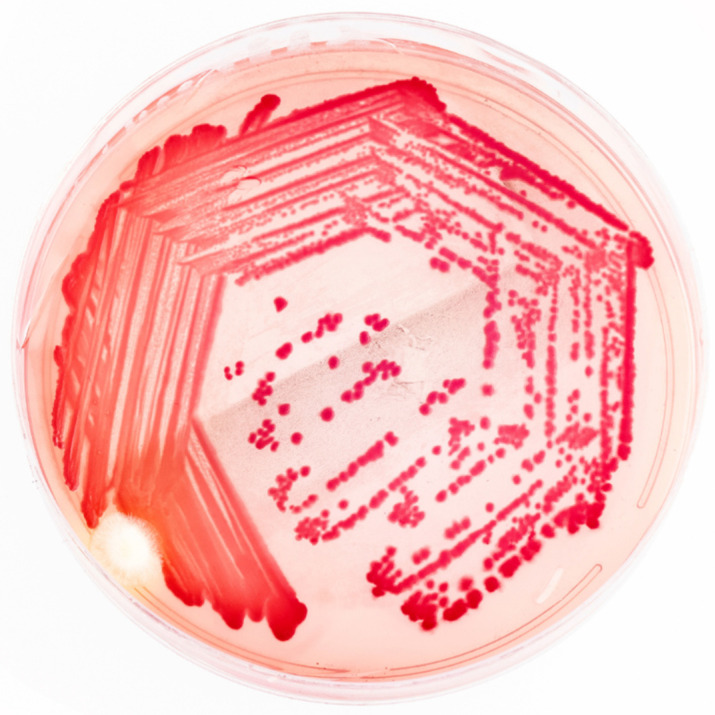
*H. pylori* can only thrive in a microaerobic environment with a nutrient-rich medium and serum. Image paid for by authors; stock photo from Canva.

**Table 1 life-12-02038-t001:** Cost Comparisons of *H. pylori* in the Southeastern United States.

Regimen	Cost
Bismuth Quadruple Therapy	$85.08
Pylera	$903.84
Helidac	$946.89
Clarithromycin Triple Therapy	$15.44
Prevpac	$224.90
Concomitant Therapy	$23.54
Levofloxacin Triple Therapy	$14.10
Rifabutin Triple Therapy	$146.33
Talicia	$697.01

**Table 2 life-12-02038-t002:** Clinical Efficacy and Safety.

Clinical Efficacy and Safety			
Author and Study	Groups Studied and Intervention	Results and Findings	Conclusions
**Study 1:** Single Capsule Bismuth Quadruple Therapy for Eradication of *H. pylori* Infection: A Real-Life Study https://pubmed.ncbi.nlm.nih.gov/33995097/ (accessed on 11 August 2022)	A total of 250 treatment-naïve patients were given esomeprazole 40 mg bid and Pylera 3 tablets QID for 10 days.	13 patients discontinued therapy due to side effects Compliance was greater than 90% Eradication rates were 90.8% 26.8% experienced adverse effects	Using a combination capsule is an effective strategy that improves compliance and yields high eradication rates with low incidence of side effects.
**Study 2:** Long-term changes in the gut microbiota after triple therapy, sequential therapy, bismuth quadruple therapy, and concomitant therapy for Helicobacter pylori eradication in Chinese children https://pubmed.ncbi.nlm.nih.gov/33899288/ (accessed on 11 August 2022)	16 patients in triple therapy group 15 patients in sequential therapy group 16 patients in bismuth quadruple therapy 16 patients in quadruple therapy Fecal samples were sampled at weeks 0, 2, 6, and 52, and 16S RNA gene sequenced to assess alterations in gut microbiota	All groups showed alterations at week 2 but were restored by week 52 Immediately following treatment in the ST group, Proteobacteria significantly increased	Transient microbiota changes may occur following *H. pylori* treatment but return to normal within a year to indicate long-term safety.
**Study 3:** Half-dose clarithromycin-containing bismuth therapy is effective and economical in treating *Helicobacter pylori* infection: A single-center, open-label, randomized trial quadruple https://pubmed.ncbi.nlm.nih.gov/30780194/ (accessed on 11 August 2022)	210 patients assigned to either half dose clarithromycin (250 mg BID) or standard dose clarithromycin (500 mg BID) containing bismuth quadruple therapy for 14 days UBT performed at 4 weeks post-treatment to assess eradication	Eradication rates were equivalent at 86.67% for both groups The standard-dose group had a higher incidence of side effects at 54.21% vs. 34.29% Lower dose also has an improved cost-effectiveness ratio	A half-dose clarithromycin regimen is as effective as the full dose regimen with improved tolerability and affordability.
**Study 4:** Efficacy of Lactoferrin with Standard Triple Therapy or Sequential Therapy for *Helicobacter pylori* Eradication: A Randomized Controlled Trial https://pubmed.ncbi.nlm.nih.gov/34609303/ (accessed on 11 August 2022)	A total of 400 *H. pylori*-infected patients were either assigned to a triple therapy or sequential therapy group, and each regimen also had a separate group that added bovine lactoferrin for 2 weeks.	Bovine lactoferrin showed improved effectiveness in the sequential therapy and triple therapy (94.5% vs. 82.8% and 85.6% vs. 70.3%).	Bovine lactoferrin could help improve eradication rates as an addition to current regimens.
**Study 5:** Efficacy and safety of twice a day, bismuth-containing quadruple therapy using high-dose tetracycline and metronidazole for second-line *Helicobacter pylori* eradication https://pubmed.ncbi.nlm.nih.gov/32074663/ (accessed on 11 August 2022)	A total of 100 patients in each group received either bismuth subcitrate 300 mg QID or BID, tetracycline 1 g, metronidazole 750 mg, and pantoprazole 20 mg BID for 1 week.	The eradication rates between the twice daily and four times daily did not differ (93.9% vs. 92.9%), while adverse effects were more common in the four times daily than twice daily (50 vs. 36).	The twice daily regimen is as effective and safe as the four-times-daily regimen.

**Table 3 life-12-02038-t003:** Comparative Studies.

Comparative Studies			
Study	Groups Studied and Intervention	Results and Findings	Conclusions
**Study 1:** High Effective of 14-Day High-Dose PPI- Bismuth-Containing Quadruple Therapy with Probiotics Supplement for *Helicobacter Pylori* Eradication: A Double Blinded-Randomized Placebo-Controlled Study https://pubmed.ncbi.nlm.nih.gov/31554388/ (accessed on 11 August 2022)	A total of 100 patients randomized to receive 7 or 14-day bismuth quadruple therapy with or without probiotic supplement.	Overall eradication rates were 68% and 96% for the 7- and 14-day probiotics group and reduced side effects compared to placebo (26% vs. 6%).	Adding probiotics can improve eradication rates and reduce side effects.
**Study 3:** Rifabutin-Based Triple Therapy or Bismuth-Based Quadruple Regimen As Rescue Therapies For *Helicobacter pylori* Infection https://pubmed.ncbi.nlm.nih.gov/32646659/ (accessed on 11 August 2022)	A total of 270 patients received rifabutin-based triple therapy for 12 days, and 153 patients received quadruple therapy with Pylera for 10 days.	Pylera therapy produced a greater eradication rate than rifabutin (88.3% vs. 61.9%).	Pylera is a good option over rifabutin, despite previous treatment failures.
**Study 4:** Two-week bismuth-containing quadruple therapy and concomitant therapy are effective first-line treatments for *Helicobacter pylori* eradication: A prospective open-label randomized trial https://pubmed.ncbi.nlm.nih.gov/31857780/ (accessed on 11 August 2022)	A total 68 patients in each group received either quadruple or concomitant therapy for 2 weeks.	The eradication rate of the quadruple therapy was higher than the concomitant group (88.2% vs. 79.4%) and had lower adverse events (33.8% vs. 51.5%).	Quadruple therapy was found to have higher eradication rates and better tolerability.
**Study 5:** Concomitant Therapy versus Triple Therapy: Efficacy in *H. pylori* Eradication and Predictors of Treatment Failure https://pubmed.ncbi.nlm.nih.gov/33645176/ (accessed on 11 August 2022)	A total 105 patients were treated with concomitant therapy and 106 with triple therapy for two weeks each.	Concomitant therapy achieved eradication rates of 91.9% and 77.2% for triple therapy.	Concomitant therapy achieved higher eradication rates than triple therapy.

## Data Availability

Data are available upon request.

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
