# Peer review of "Helicobacter Pylori: A Review of Current Treatment Options in Clinical Practice"

_life, 2022, doi:10.3390/life12122038_

Round 1
Reviewer 1 Report
Journal: Life
Title of the manuscript: Helicobacter Pylori: A Review of Current Treatment Options in
Clinical Practice
Manuscript ID: life-2039105
In general manuscript provide useful information in the form of review about Helicobacter Pylori and its treatment options in clinical practice. The review article delivered the significant information, it is highly useful for the readers of this particular field. As over all the manuscript is well designed and discussed well.
However, in my opinion if authors provide some more Figures to this review article, it may provide better understanding of the readers.
Author Response
Date: Nov 2022
Manuscript number: life-2039105
Title: Helicobacter Pylori: A Review of Current Treatment Options in Clinical Practice
Dear Editor in Chief,
Thank you for your recent correspondence regarding our paper on Helicobacter Pylori: A Review of Current Treatment Options in Clinical Practice, submitted to Life. Here is a corrected manuscript with a point by point response to comments from the Reviewers.
Specifically, with regard to Reviewers' Comments to Author:
Title of the manuscript: Helicobacter Pylori: A Review of Current Treatment Options in
Clinical Practice
Manuscript ID: life-2039105
Reviewer 1
In general manuscript provide useful information in the form of review about Helicobacter Pylori and its treatment options in clinical practice. The review article delivered the significant information, it is highly useful for the readers of this particular field. As over all the manuscript is well designed and discussed well.
However, in my opinion if authors provide some more Figures to this review article, it may provide better understanding of the readers.
We appreciate this comment from the reviewer, and we have added more figures to the paper. These figures are stock images that are paid for by the authors who have a subscription to Canva. Please see the Free Media License Agreement: https://www.canva.com/policies/free-media-license-agreement-2022-01-03/
Figure 2. H.Pylori bacterium, colored transmission electron micrograph (TEM). Image paid for by authors, stock photo from Canva.
Figure 3. Stained microscopic image of stomach biopsy, reactive epithelium from H. Pylori-induced gastritis. Image paid for by authors, stock photo from Canva.
Figure 4. H. Pylori can only thrive in a microaerobic environment with a nutrient-rich medium and serum. Image paid for by authors, stock photo from Canva.
Reviewer 2
My comments are as follows:
- There are some mistakes of grammar and format. For example, in page 2, line 82, the expression of “A 2018 Cochrane review of non-invasive diagnostic tests 82 for H. pylori found that 83 urea breath tests had high diagnostic accuracy, while serology and stool antigen tests had 84 lower accuracy in detecting H. pylori infection” should be modified. And the document format of the references needs to be modified according to the requirements of the journal. This has been fixed.
- Please provide an overview and detailed description of Helicobacter pylori which it is very important to understand the determination of treatment plan, such as the discovery, research progress and pathogenesis. We have added this to the manuscript.
- H. Pylori Overview
- Pylori was discovered as a spiral bacteria in the stomach of dogs by Giulio Bizzozero in 1892. Since they are Campylobacter-like spiral bacteria, Barry Marshall and Robin Warren dubbed them Campylobacter pyloridis in 1983. Goodwin et al. designated it Helicobacter pylori in 1989 because of its helical form and prevalence in the pyloric area of the stomach. H. Pylori is a 0.5–1 m wide, 2–4 m long, S-shaped, short helical, Gram-negative bacteria that infects more than half of the world's population [10].
- Pylori is a short helical, S-shaped, Gram-negative bacteria measuring 0.5–1 m in width and 2–4 m in length. It is particularly prevalent in the pyloric area of the stomach, where it causes persistent gastric infection. It is believed that more than half of the world's population is infected with these bacteria. The specific mode of transmission and infection of H. Pylori is yet unknown, but the feces-to-mouth and mouth-to-mouth pathways via water or food consumption are believed to be quite common [10]. Figures 2 & 3.
The severity of H. Pylori-caused gastric atrophy and gastric cancer has been the subject of growing study attention over the past three decades, particularly in terms of pathogenicity, microbial activity, genetic predisposition, and clinical therapies. Studies have revealed a connection between H. Pylori infection and malabsorption of important micronutrients, and H. Pylori infection may influence the prevalence of malnutrition in some high-risk populations. On the other hand, dietary factors may play a significant role in H. Pylori infection, and it has been reported that an adequate and balanced diet, particularly a high consumption of fruits and vegetables and low consumption of processed salty foods, has a protective effect against the consequences of H. Pylori infection [10]. Figure 4.
The combination of drug loading of nano materials and natural drug loading may have a better antibacterial effect in future clinical studies. Biotherapeutics derived from microorganisms are crucial for combating infections such as H. Pylori, but these biotherapeutics must be alive at the time of administration to be effective. Numerous potentially therapeutic species are anaerobes and, as a result, their generation is almost impossible due to the low efficacy of present protective techniques.
Inspired by the features of cells in living animals, these new hybrids comprised of living cells and abiotic materials with diverse forms and functions can increase cell stability and allow for the introduction of novel activities into living cells. Numerous applications, such as bioelectronics, cell protection, cell treatment, and biocatalysis, have a significant deal of potential for single-cell nanoshells [11].
For example, Bacteroides thetaiotaomicron is a self-assembling cellular coating [12]. Even in the absence of conventional cryoprotectants, this coating exhibits resistance to harsh processing conditions and oxygen exposure. This innovation will expand the variety of microorganisms that can be produced in a stable manner and promote the development of new strains of interest by ensuring their survival after manufacturing. Reversible cell encapsulation, made possible by the presence or absence of glucose is another emerging development that significantly increases cell survival in a variety of hostile environments without interfering with the cells’ natural growth [13].
- Etiology, Epidemiology, Pathophysiology
- Pylori can be transmitted by the fecal-oral, gastric-oral, oral-oral, and sexual pathways. Lower socioeconomic status is a significant risk factor for a higher infection prevalence [14]. The prevalence of H. Pylori varies across the globe, with a 5% prevalence in children younger than ten years old in the United States. The Hispanic and African American populations are more prevalent than the White population [14].
In H. Pylori infection, four key factors contribute to the development of clinical illnesses such as gastritis and ulcer. First, the urease activity of H. Pylori serves a crucial function in neutralizing the stomach's acidic environment. Second, the H. Pylori bacterium moves toward the host gastric epithelial cells via flagella-mediated motility. The subsequent interaction between bacterial adhesins and host cell receptors results in effective colonization and sustained infection. In addition, H. Pylori produce numerous effector proteins/toxins, such as cytotoxin-associated gene A (Cag A) and vacuolating cytotoxin A (VacA), that cause host tissue damage. H. Pylori gastritis is characterized by both acute and chronic inflammation due to the stimulation of eosinophils, neutrophils, mast cells, and dendritic cells. Additionally, the gastric epithelial layer secretes chemokines to trigger innate immunity and activates neutrophils, which further damage the host tissue, resulting in the establishment of gastritis and ulcer [14].
- The combination of drug loading of nano materials and natural drug loading may have better antibacterial effect in future clinical studies. More references should be added in the introduction to learn more about development of single cell technology. (Please check Nanoscale, 2018, 10, 3112-3129,ACS Nano, 2019, 13, 14459-14467, J. Am. Chem. Soc. 2022, 144, 2438-2443). We thank the reviewer for this helpful and interesting information. We have added this section to the manuscript.
- H. Pylori Overview
- Pylori was discovered as a spiral bacteria in the stomach of dogs by Giulio Bizzozero in 1892. Since they are Campylobacter-like spiral bacteria, Barry Marshall and Robin Warren dubbed them Campylobacter pyloridis in 1983. Goodwin et al. designated it Helicobacter pylori in 1989 because of its helical form and prevalence in the pyloric area of the stomach. H. Pylori is a 0.5–1 m wide, 2–4 m long, S-shaped, short helical, Gram-negative bacteria that infects more than half of the world's population [10].
- Pylori is a short helical, S-shaped, Gram-negative bacteria measuring 0.5–1 m in width and 2–4 m in length. It is particularly prevalent in the pyloric area of the stomach, where it causes persistent gastric infection. It is believed that more than half of the world's population is infected with these bacteria. The specific mode of transmission and infection of H. Pylori is yet unknown, but the feces-to-mouth and mouth-to-mouth pathways via water or food consumption are believed to be quite common [10]. Figures 2 & 3.
The severity of H. Pylori-caused gastric atrophy and gastric cancer has been the subject of growing study attention over the past three decades, particularly in terms of pathogenicity, microbial activity, genetic predisposition, and clinical therapies. Studies have revealed a connection between H. Pylori infection and malabsorption of important micronutrients, and H. Pylori infection may influence the prevalence of malnutrition in some high-risk populations. On the other hand, dietary factors may play a significant role in H. Pylori infection, and it has been reported that an adequate and balanced diet, particularly a high consumption of fruits and vegetables and low consumption of processed salty foods, has a protective effect against the consequences of H. Pylori infection [10]. Figure 4.
The combination of drug loading of nano materials and natural drug loading may have a better antibacterial effect in future clinical studies. Biotherapeutics derived from microorganisms are crucial for combating infections such as H. Pylori, but these biotherapeutics must be alive at the time of administration to be effective. Numerous potentially therapeutic species are anaerobes and, as a result, their generation is almost impossible due to the low efficacy of present protective techniques.
Inspired by the features of cells in living animals, these new hybrids comprised of living cells and abiotic materials with diverse forms and functions can increase cell stability and allow for the introduction of novel activities into living cells. Numerous applications, such as bioelectronics, cell protection, cell treatment, and biocatalysis, have a significant deal of potential for single-cell nanoshells [11].
For example, Bacteroides thetaiotaomicron is a self-assembling cellular coating [12]. Even in the absence of conventional cryoprotectants, this coating exhibits resistance to harsh processing conditions and oxygen exposure. This innovation will expand the variety of microorganisms that can be produced in a stable manner and promote the development of new strains of interest by ensuring their survival after manufacturing. Reversible cell encapsulation, made possible by the presence or absence of glucose is another emerging development that significantly increases cell survival in a variety of hostile environments without interfering with the cells’ natural growth [13].
Sincerely,
Dr. Elyse M. Cornett, PhD
Assistant Professor, Department of Anesthesiology
Director of Research, Department of Anesthesiology
Assistant Professor, Department of Pharmacology, Toxicology & Neuroscience
Assistant Professor of Research, Department of Anesthesiology, LSU New Orleans
LSU Health Shreveport
1501 Kings Highway
P.O. Box 33932
Shreveport, LA 71130-3932
c.248-515-9211
w.318-626-2348
Sincerely,
Alan David Kaye, MD PhD
Professor, Louisiana State University Health Sciences Center, Shreveport, Departments of Anesthesiology and Pharmacology, Toxicology and Neurosciences, Pain Fellowship Program Director, Provost, Vice Chancellor of Academic Affairs, Chief Academic Officer, alan.kaye@lsuhs.edu

Reviewer 2 Report
My comments are as follows:
1. There are some mistakes of grammar and format. For example, in page 2, line 82, the expression of “A 2018 Cochrane review of non-invasive diagnostic tests 82 for H. pylori found that 83 urea breath tests had high diagnostic accuracy, while serology and stool antigen tests had 84 lower accuracy in detecting H. pylori infection” should be modified. And the document format of the references needs to be modified according to the requirements of the journal.
2. Please provide an overview and detailed description of Helicobacter pylori which it is very important to understand the determination of treatment plan, such as the discovery, research progress and pathogenesis.
3. The combination of drug loading of nano materials and natural drug loading may have better antibacterial effect in future clinical studies. More references should be added in the introduction to learn more about development of single cell technology. (Please check Nanoscale, 2018, 10, 3112-3129, ACS Nano, 2019, 13, 14459-14467, J. Am. Chem. Soc. 2022, 144, 2438-2443).
Author Response

(The authors gave the same response as above.)

Round 2
Reviewer 2 Report
The authors have modified the manuscript properly that I recommend to accept it.